

# Research on the prediction of English topic richness in the context of multimedia data

Jie Jiao[1] and Hanan Aljuaid[2]

[1] Jiaozuo Normal College, Jiaozuo, China
[2] Computer Sciences Department, College of Computer and Information Sciences, Princess Nourah bint Abdulrahman University (PNU), Riyadh, Saudi Arabia

## ABSTRACT

With the evolution of the Internet and multimedia technologies, delving deep into multimedia data for predicting topic richness holds significant practical implications in public opinion monitoring and data discourse power competition. This study introduces an algorithm for predicting English topic richness based on the Transformer model, applied specifically to the Twitter platform. Initially, relevant data is organized and extracted following an analysis of Twitter's characteristics. Subsequently, a feature fusion approach is employed to mine, extract, and construct features from Twitter blogs and users, encompassing blog features, topic features, and user features, which are amalgamated into multimodal features. Lastly, the combined features undergo training and learning using the Transformer model. Through experimentation on the Twitter topic richness dataset, our algorithm achieves an accuracy of 82.3%, affirming the efficacy and superior performance of the proposed approach.

## INTRODUCTION

With the popularization of the Internet and the promotion of media convergence construction, the topic richness prediction for mainstream social media is a high-profile research topic in the era of all media, which can be widely used in the field of public opinion monitoring and data discourse power competition, and has considerable practical significance (*Wu et al., 2017*). Twitter is a mainstream social media with wide influence. The research on its topic richness prediction helps calculate the future popularity of information, find out hot topics and extract the rules of information dissemination, which can be further used in information retrieval, public opinion analysis and enterprise marketing (*Wu, Chen & Jiang, 2017*).

Topic richness prediction (*Ai et al., 2020*) refers to the prediction of the future attention degree of the information posted by users. However, the definition of richness often depends on the platform of social media, and different network platforms have different evaluation metrics. Many current studies only use a single evaluation token. For example, *Pinto, Almeida & Goncalves (2013)* define richness as the number of views of online videos on YouTube and propose a method to predict the number of views at specified future

Corresponding author
Jie Jiao, 1295011005@jzsz.edu.cn

moments by training the multivariate linear model (ML model) and multivariate radial basis functions model (MRBF model). *Hong, Dan & Davison (2011)* regard the number of Twitter retweets at a given moment as the topic richness of Twitter. To evaluate topic richness as more representative and universal, this article defines the sum of the number of retweets, comments and favourites of Twitter as the interaction value, which is used as the standard to measure the topic richness of Twitter.

At present, the mainstream method of topic richness prediction in social media is feature-based model prediction. *Wu et al. (2016)* investigate the Flickr platform when studying the topic richness of social media and believe that the spatial and temporal information of photos and posts on the Flickr platform are very important for the final richness. *Mazloom, Pappi & Worring (2018)* conducted a study on Instagram posts and found that the classes of the posts can benefit from accurately predicting the richness. *Vilares, Alonso & Gómez-Rodríguez (2015)* pay more attention to the textual features and the richness prediction based on lexical and syntactic processes when studying the information on Twitter.

These methodologies, rooted in the unique characteristics of social media, underscore the critical role of feature extraction in social media analysis. However, it becomes apparent that the approaches mentioned so far are primarily focused on individual categories, lacking a holistic consideration of various influencing factors and not fully exploiting the multimodal nature of social media data. To address this limitation, there is a growing need for more comprehensive feature extraction methods that can capture the intricate interplay of diverse elements within the social media landscape. Multimodal features, encompassing text, images, and potentially other modalities present an opportunity to enrich the understanding of social media content. By leveraging the synergies between these modalities, a more nuanced and complete representation of topics can be achieved. Our ongoing research endeavours to bridge this gap by exploring advanced feature extraction techniques that embrace the multimodal nature of social media data. Through the integration of diverse features and a more comprehensive consideration of influencing factors, we aim to enhance the accuracy and richness of English topic representation and prediction. This holistic approach reflects the complex and dynamic nature of social media discussions, providing a more robust foundation for predictive models in this evolving landscape. Therefore, we propose an English topic richness prediction algorithm based on an improved Transformer.

Firstly, based on the analysis of the data, features are extracted from three aspects: blog information, topic information and user information. Among the blog features, the content features and time features are constructed, and the topic features are derived based on the blog features. For user features, the influence of users is visualized, and the rank distribution of users is calculated proportionally from the perspective of statistics as new user features. The algorithm in our article adopts the classification framework. After the fusion of multi-class features, the rank of topic richness can be divided. The transformer model is used as the classifier to predict the topic richness of Twitter, which can transform

the topic richness prediction into a classification implement. The main contributions are as follows:

1) We propose the method of feature extraction and fusion, which contains the blog feature, topic popularity feature and user feature.
2) We suggest a transformer-based encoder-decoder framework to achieve the prediction the topic richness.
3) We achieve the best performance while comparing with the competitive methods.

## RELATED WORK

In this article, feature extraction and feature fusion are used to quantify the number of retweets, comments and favourites on Twitter. Feature extraction is widely used in pattern recognition and machine learning. The quality of feature extraction is closely related to the generalization ability. Feature fusion refers to the optimization and combination of different feature vectors extracted from the same pattern (*Xie et al., 2022*). According to the different phases, feature fusion can be divided into two categories: one is early fusion, which is to fuse different features before model training and exploit fused features for training and testing by concatenation and addition; another is late fusion, in which the model is trained before completely fusing and the fusion is improved after several training sessions, such as Single Shot Multibox Detector (SSD) (*Liu et al., 2016*), Multi-scale CNN (MS-CNN) (*Cai et al., 2016*) and Feature Pyramid Network (FPN) (*Lin et al., 2017*). The proposed algorithm adopts the concatenated method in the early fusion.

The topic richness prediction of social media also depends on the construction of models. Transformer is a deep learning model proposed in recent years, which is widely used in the classification field. We regard the transformer as a classifier to fit the residual between the predicted value and the real value in the previous iteration before fine-tuning the model. In the whole iteration process, a loss function should be defined to make the predicted value as close to the true value as possible and to ensure a large generalization ability. Finally, we provide the test information for the trained model to obtain the prediction results.

In the following comparative experiments, the deep neural networks (DNN) are applied to design the topic richness prediction methods based on the deep learning framework, whose performance is compared with the algorithm in this article. Based on the extension of perceptron, DNN can be regarded as a neural network with multiple hidden layers, which can be divided into three types: input layer, hidden layer and output layer. DNN network includes two hidden layers. The beginning layer is the input layer, which can input the fused features. The middle two layers are hidden layers, which are often 256 and 128 dimensions, respectively. The final output layer is a 1-dimensional output.

When predicting the richness of social media topics, user comments often encompass various aspects, necessitating the consideration of global information. Social media data is characterized by its timeliness and diversity, as new topics continually emerge while old

ones gradually wane. Feature fusion enables models to concurrently incorporate information from diverse features, offering more comprehensive inputs. This approach enhances the model's ability to comprehend multimodal data in social media, including text, images, and more. Nevertheless, conventional methods face challenges in accurately representing complex English topic environments. Our research endeavours to overcome these challenges by integrating deep learning models, specifically employing techniques like feature fusion and transformer. This integration aims to achieve precise representation and prediction of rich English topics, enhancing the overall effectiveness of the predictive model. In this article, a transformer (*Vaswani, Shazeer & Parmar, 2017*) is used to train multimodal features of Twitter information. Based on fully mining and constructing effective features, machine learning models are used to improve the performance of the algorithm.

# PREDICTION MODEL OF ENGLISH TOPIC RICHNESS BASED ON TRANSFORMER

In this article, we propose a Twitter topic richness prediction algorithm based on the transformer. It mainly includes two modules: feature extraction and fusion and richness classification model based on transformer, as shown in Fig. 1.

This algorithm is primarily composed of two key modules: Feature Extraction and Fusion and a Transformer-based Richness Classification model. In the Feature Extraction and Fusion module, our focus is on extracting and integrating diverse features from Twitter data to comprehensively capture the diversity and complexity of topics. The fusion of features from different dimensions forms a comprehensive feature vector, providing richer inputs for subsequent models. In the Transformer-based Richness Classification Model, we construct a classification model based on the transformer. This model learns the mapping from fused features to topic richness, enabling accurate predictions of topics. Leveraging the sequence modelling capabilities of the transformer, we better capture long-distance dependencies in blog text. Adjustments to attention mechanisms are made to accommodate the handling of multimodal inputs, enhancing the model's focus on features.

Additionally, we introduce user features twice into the model, first through an intermediate fusion after the Transformer layer and secondly through a final fusion before the ultimate output. Through the collaborative interaction of these two key modules, our algorithm comprehensively considers blog content, user behaviour, and time features to accurately predict the richness of Twitter topics. The introduction of this method aims to provide deeper insights into social media analysis, promoting a more comprehensive understanding of topic diversity.

## Feature extraction and fusion

The quantification of topic richness on Twitter can be achieved by examining the metrics such as the number of retweets, comments, and favourites associated with the content. In light of this, our approach in this article involves breaking down topic richness into three key dimensions: blog posts, users, and popularity. We then perform feature extraction and

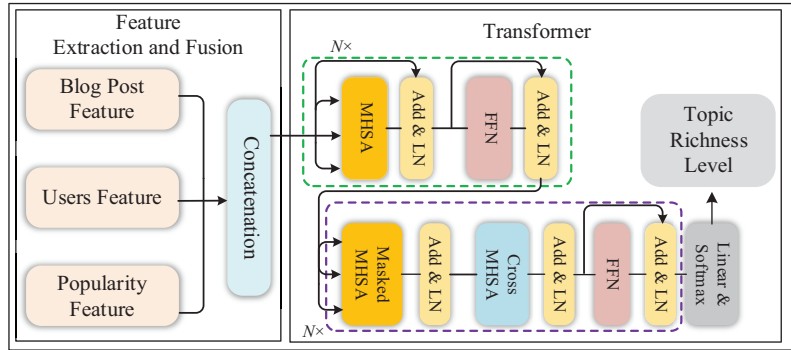

**Figure 1** Framework of Twitter topic richness prediction algorithm based on Transformer.

**Table 1** Blog post features.

| Names | Descriptor | Sources |
| --- | --- | --- |
| Content | Content of blog post | Primary |
| Time | Time of blog present | Primary |
| Title | Title of blog post | Primary |
| Tags | Topic numbers of blog post | Extracted |
| Content length | Length of blog post | Extracted |
| Video | Video exist | Extracted |
| At | Association with others | Extracted |

multimodal fusion to comprehensively capture the diverse aspects contributing to the richness of topics in the Twitter ecosystem.

### Blog feature extraction

We can extract and construct the blog features as shown in Table 1 by further analysis of the original text content and publication time of the blog. Considering that a blog contains a body, title and topic, which are the important content features, we perform the data cleaning for the original blog, whose structure is neat and format is relatively uniform. After the first data cleaning of the original blog, we can achieve the initial title, topic and text. In the initial text, some of the blog posts have special symbols @ and URLs which indicate the association with other users and the existence of video links, respectively. Based on this, we perform the second clean for the text of the first edition to obtain the final text and additional features of the Boolean type of whether there is video or not and whether it is associated with other users, where one means yes and 0 means no. The multimodal features, such as text, title and topic, cannot meet the requirements of model training. According to this, two new features, text length and topic number, are obtained by further numerical construction. For the post-release date, we counted the peak and low peak of social media traffic at different periods of the day, as shown in Figs. 2 and 3, where the horizontal axis for 24 h a day in different periods and the vertical axis represents the
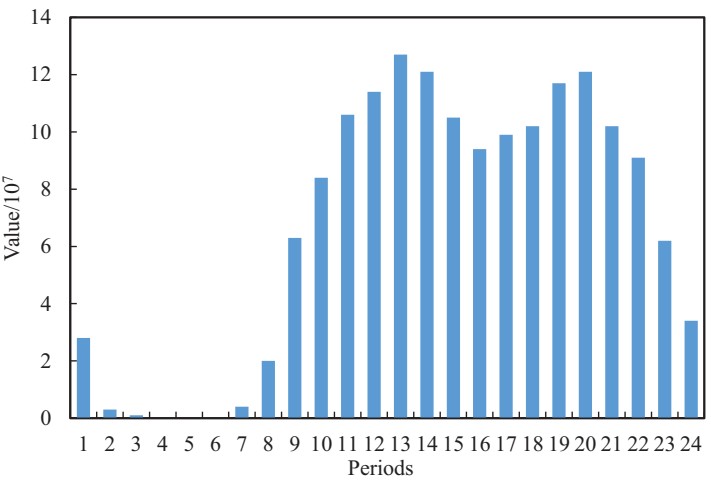

**Figure 2 Total interaction value of the posts in each time period.**

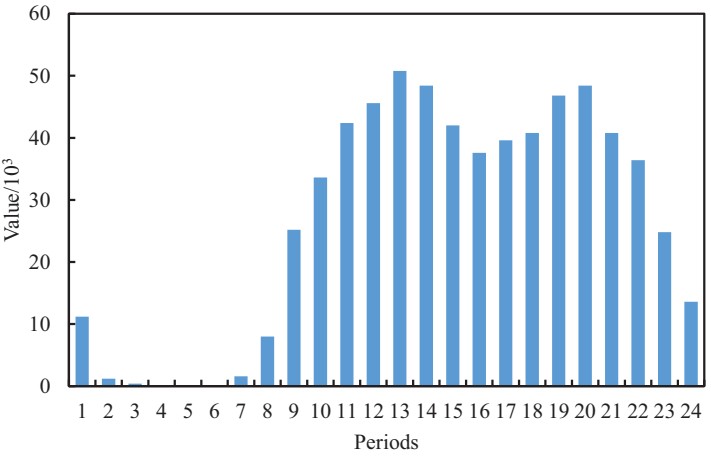

**Figure 3 Average interaction value of the posts in each time period.**

total interaction value and the interaction of the average value in the dataset's all the posts for a day. It can be seen that different periods occupy different interactive values, which reflects the richness of time sensitivity.

### Topic popularity feature extraction

The topic popularity feature is a new feature derived from the topics in blog features, which mainly reflects the influence of a topic, namely the topic index. As shown in Table 2, we mainly construct three topic features: the number of blogs involved in the topic (IndexCount), the total interaction value of the topic (IndexTotal) and the average interaction value of the topic (IndexAvg). The interaction value is the sum of retweets, favourites and comments of blog posts, the total interaction value is the sum of retweets, favourites and comments of all blog posts related to the topic, and the average interaction

**Table 2 Topic popularity features.**

| Names | Descriptor | Sources |
|---|---|---|
| $Index_{Count}$ | Number of blogs involved | Extracted |
| $Index_{Total}$ | Total interaction value | Extracted |
| $Index_{Avg}$ | Average interaction value | Extracted |

value is the sum of retweets, favourites and comments of a single blog post related to the topic.

### User feature extraction

User features are extracted and constructed from user portrait information. The original user profile features, such as the number of posts, followers and favourites, can roughly reflect the user's influence. However, they are not detailed and comprehensive enough. On this basis, we further visualize the influence of users, which can be mainly regarded as the average number of reposts (RepostAvg), average number of comments (CommentAvg), the average number of favourites (CollectAvg) and average interaction value (TotalAvg) which are four newly constructed user features. The relationships between them are as follows:

$$Total_{Avg} = Repost_{Avg} + Comment_{Avg} + Collect_{Avg}. \tag{1}$$

In addition, we perform calculations from the perspective of statistics for the probability of users' blog posts in different ranks. The rank divisions are presented in Table 3.

The proportion of users' blog posts in different ranks is calculated using the following formula:

$$P_i = \frac{Count_i}{All} \tag{2}$$

where $Count_i$ denotes the number of blog posts in the i-th rank.

### Feature fusion

The last step of feature processing is feature fusion, as shown in Fig. 1, which will extract the calculated blog features fpost, topic feature ftag and user features fuser and concatenate themselves to obtain the final multimodal fusion features fall, and their relation is shown as follows:

$$f_{all} = \left[ f_{post}, \ f_{tag}, \ f_{user} \right] \tag{3}$$

$$f_{tag} = Sigmoid \left( Index_{Avg} \right) \tag{4}$$

$$f_{user} = Sigmoid \left( Total_{Avg} \right). \tag{5}$$

In summary, feature processing encompasses the exploration, extraction, and construction of blog, topic, and user features. Initially, a thorough analysis of blog text content involves leveraging natural language processing to extract keywords, themes,

**Table 3 Division of gears.**

| Gears | $Total_{Avg}$ | Weight |
|---|---|---|
| 1 | 0–50 | 1 |
| 2 | 50–200 | 50 |
| 3 | 200–500 | 100 |
| 4 | >500 | 200 |

emotions, and other relevant information. Simultaneously, time features like hours and days of the week are extracted from the blog post timestamps to account for the temporal impact on topic richness. Following this, employing techniques such as correlation analysis helps identify the topics and keywords within the blog, leading to the creation of a topic vocabulary. Quantifying the frequency of each topic occurrence in a blog serves as a feature reflecting the topic's significance. Calculating the correlation between different topics within a blog captures the interrelatedness and forms the primary characteristics. Subsequently, user behaviour analysis encompasses parameters like posting frequency, the breadth of discussed topics, and activity levels. Considering the user's social connections and following relationships aids in reflecting their social influence. User features are derived from their prior blog interaction records. Through feature fusion, features from various dimensions are combined into a comprehensive feature vector, offering a more holistic representation of the intricate relationships between blogs, topics, and users.

## Transformer encoder-decoder framework

The transformer can be divided into two parts: encoder and decoder, as shown in Fig. 1. The basic structures of the encoder and decoder are a multi-head self-attention layer and a feedforward neural network layer. Besides, the residual structure between sub-blocks is formed to improve the elasticity of the network, where N is the number of blocks.

Given input multimodal fused feature fall, the basic unit calculation process of the Transformer framework is as follows:

Firstly, the basic unit, the self-attention mechanism, of the transformer is calculated. The multimodal features fall are linearized *via* the weight matrix $W^Q$, $W^K$, $W^V$ separately, which can strengthen the complexity of the features and enhance the malleability of the features, as follows:

$$Q = f_{all}W^Q \tag{6}$$
$$K = f_{all}W^K \tag{7}$$
$$V = f_{all}W^V. \tag{8}$$

Calculate the output features of the self-attention layer Z, where $d_k$ is the dimension of the key vector.

$$Z = \text{Softmax}\left(\frac{QK^T}{\sqrt{d_k}}\right)V. \tag{9}$$

Then, the output attended features $Z_{mul}$ of the multi-heads attention layer is calculated, where H is the number of attention heads, $Z_i$ represents the i-th attention head, [...,...] function represents the concatenation of all heads and $W^O$ is the additional weight matrix. Note that $Z_{mul}$ and fall has the same dimension.

$$Z_{mul} = [Z_0, Z_1, \ldots, Z_i]W^O. \qquad (10)$$

Perform summation and layer normalization, where the LN function represents layer normalization.

$$Z_{mul} = LN(Z_{mul} + f_{all}). \qquad (11)$$

Finally, feed $Z_{mul}$ into the feedforward neural network, and then perform the summation and layer normalization again.

The algorithm we propose can be widely applied in the real world, especially in showcasing individual preferences and public attention on social media and assisting in predicting societal opinion trends. Firstly, through the analysis of individual behaviours on social media, our algorithm can capture users' personal preferences. It can recommend blog content that aligns more with users' interests and preferences based on their past interactions and posting history, enhancing user experience. Secondly, the algorithm not only focuses on individual users but also analyzes overall topics and trends on social media, showcasing the direction of public attention. By analyzing a large volume of blogs, the algorithm can identify current hot topics on social media, helping users understand the focal points of public opinion. It can also assess the activity levels of different topics, revealing the extent of public attention on specific subjects and aiding social media platforms in adjusting content strategies. Finally, through in-depth analysis of social media data, our algorithm contributes to predicting societal opinion trends. By considering blog content, user behaviour, and time features, it accurately predicts the richness of specific topics, enabling users to understand the popularity of different topics on social media. The integration of these functionalities makes our algorithm applicable not only for personalized experiences for individual users but also provides deeper insights for social media platforms and opinion analysts, supporting more intelligent and targeted decision-making.

## EXPERIMENT AND ANALYSIS

### Dataset

The dataset of Twitter topic richness prediction (https://zenodo.org/records/7139621) is composed of 500 randomly selected users and contains a total of one million original blog posts published before the sampling date by these 500 users. In the experiment, 90% random blog content data of each user is used to form the training set, and the remaining 10% data of each user is used as the test set. In the training process, 80% of the training set is randomly selected for model training, and the remaining 20% is used for algorithm verification.

**Table 4 Experimental parameter settings.**

| parameter | The values |
|---|---|
| Number of codec layers | 6 |
| Attention head count | 8 |
| Hide the layer dimension | 1,024 |
| Dropout | 0.3 |
| Model optimizer | Adam |
| Maximum sequence length | 16 |
| Batch_size | 50 |

The user data in the dataset includes user Id, user profile, number of all blog posts, number of followers, number of favourites and extraction date. The original blog data in the training set includes post Id, user Id, text content of the blog, publication time of the blog, the number of retweets, comments and favorites of the blog on the extraction date. The retweets, comments and favourites of the test set are not disclosed.

**Implement details**

The parameters of the Transformer model in this experiment are shown in Table 4. The incorporation of fused features into the Transformer model is executed through the following steps. Initially, we conduct the mining, extraction, and construction of fused features for the data, with each feature set containing pertinent information regarding blogs, topics, and users. To ensure effective model training, we standardize or normalize the features, aligning them on a consistent scale. Next, we adapt the architecture of the Transformer model to seamlessly incorporate the input from fused features. This adjustment may involve fine-tuning attention mechanisms to adeptly handle multimodal inputs.

Additionally, we introduce extra neural network layers to the model, enhancing its capacity to leverage the information embedded in fused features. Through these meticulously executed steps, the fused features seamlessly integrate into the Transformer model. This integration furnishes the model with a more intricate and comprehensive set of input information, consequently augmenting its predictive capabilities for assessing topic richness.

The evaluation ranks of the experiment are divided into four levels according to the classification rule. The ranks of interaction value (the sum of retweets, comments and favourites) are the first rank (0–50), the second rank (50–200), the third rank (200–500), and the fourth rank (more than 500). The weights corresponding to ranks are shown in Table 3.

Following this classification rule, we can compute the accuracy of the interaction value predicted from each blog post. The calculation formula for accuracy is as follows:

$$\text{Accuracy} = \frac{\sum_{i=1}^{4} \text{Weight}_i \times \text{Count\_r}_i}{\sum_{i=1}^{4} \text{Weight}_i \times \text{Count}_i} \tag{12}$$

**Table 5 Comparative experiments with methods based on deep learning framework.**

| Methods | Type of the data process | Accuracy (%) |
| --- | --- | --- |
| RNN | Feature standardize | 39.1 |
| BiRNN | Dropout invalid data | 45.2 |
| GRU | Label normalization | 59.1 |
| BiGRU | Label normalization | 68.6 |
| LSTM | Classify users | 61.9 |
| BiLSTM | Classify users and label normalization | 74.9 |
| Roformer | Classify users and label normalization | 77.4 |
| Mixformerv2 | Classify users and label normalization | 76.9 |
| TTST | Classify users and label normalization | 79.9 |
| Ours | | 82.3 |

where $Weight_i$ is the weight of the $i^{th}$ rank, $Count\_r_i$ refers to the number of correct predicted blog posts in the $i^{th}$ rank, and $Count_i$ is the number of posts in the $i^{th}$ rank.

## Results and Discussion

### Comparison results of different models

In order to verify the effect of the proposed model, several classical topic richness prediction models are selected for experimental comparison, such as RNN (*Zaremba, Sutskever & Vinyals, 2014*), BiRNN, GRU, BiGRU, LSTM (*Xu et al., 2024*), BiLSTM, Roformer (*Su et al., 2024*), Mixformerv2 (*Cui et al., 2024*) and TTST (*Xiao et al., 2024*) on the same dataset.

The experimental results highlight the effectiveness of the proposed English topic richness prediction model based on Transformer in Table 5, achieving an accuracy of 82.3% on the Twitter dataset. This marks a substantial improvement in performance compared to other models. Notably, when compared to the original RNN and BiRNN, the proposed method demonstrates impressive enhancements of 43.2% and 37.1%, respectively. This can be attributed to the superior feature performance and model structure extracted in this study. In the case of GRU and LSTM, the proposed method shows improvements of 23.2% and 20.4%, respectively. This improvement is due to the more efficient and accurate structures of these methods compared to the traditional RNN. Additionally, the bidirectional cyclic networks BiGRU and BiLSTM benefit from their unique decoding methods, resulting in a significant performance boost. However, they still fall short compared to the method introduced in this article. Comparing our method to Roformer, we observe a noteworthy 4.9% improvement in accuracy, emphasizing the superior handling of sequence data by our model, particularly in capturing long-distance dependencies in text. In the case of Mixformerv2 and TTST, our proposed methods achieve accuracy improvements of 5.4% and 2.4%, respectively. Finally, the approach presented in this article maximizes the distinct characteristics of multimodal data at the feature level. It takes into account three crucial aspects: blog post, user, and popularity, integrating them into the interaction value feature. This comprehensive consideration

| Table 6 Comparative experiments with user matching methods. | | |
|---|---|---|
| **Methods** | **Type of the methods** | **Accuracy (%)** |
| User matching | Max rank of user | 75.1 |
| | Max rank with corresponding period | 78.2 |
| Ours | | 82.3 |

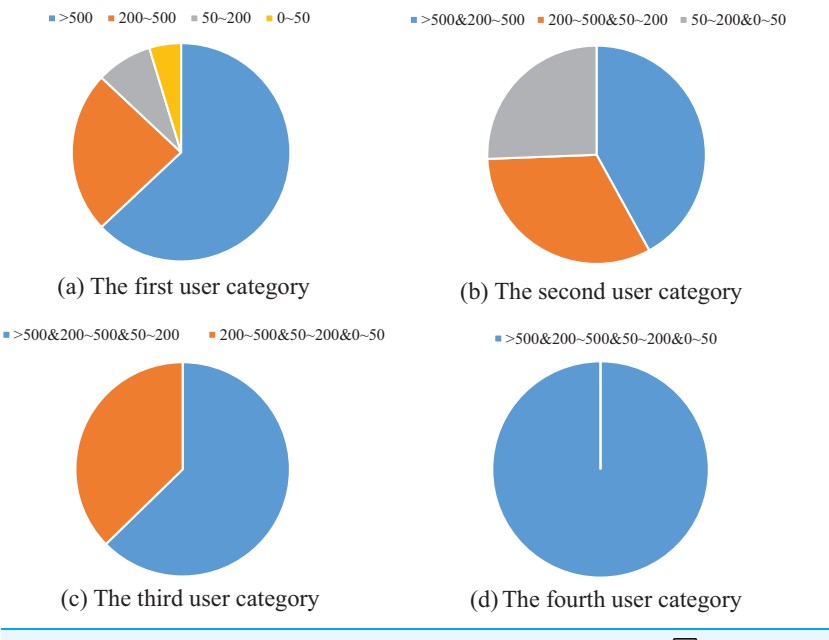

(a) The first user category

(b) The second user category

(c) The third user category

(d) The fourth user category

**Figure 4 Example of user classification.**

yields the best experimental results, emphasizing the robustness and effectiveness of our proposed model in capturing the richness of English topics in social media discussions.

### Comparison with user matching method

The user matching method is a method based on user features that do not depend on any training. Such a method relies on the statistical information of users' blog richness, which can closely associate the blog topic richness with users. The user matching method can be divided into two methods: (a) matching method of the max rank of the user and (b) matching method of maximum weight rank according to the user's corresponding period.

The user matching method by the max rank of the user is that calculate the weight distribution of each rank of the user's blog and multiply the number of blog posts by the corresponding weight. The user matching method by the maximum weight rank in the corresponding period is firstly to count the number of posts in each period of the day to obtain the rank with the maximum weight, then match them by the user and time. As shown in Table 6, the experimental results show that the accuracy of the user-matching method can reach 78.2%. Although the user matching method is very comprehensive in the mining of user features, it still does not exceed our method in terms of accuracy.

### Blog post rank classification results

According to the above ranks, we classify the users and blog posts with fixed weights. The classification is based on the following:

User category 1: More than 80% of the posts belong to a certain segment of users (1, 2, 3, 4);

User category 2: More than 80% of blog posts belong to two adjacent segments (12, 23, 34);

User category 3: Users whose blog posts are in three adjacent segments (123, 234);

User category 4: Remaining users.

The classification results are shown in Fig. 4. It can be seen from Fig. 4A that most of the users with fixed posts are in the third and fourth categories, which are the users with high topic richness. As shown in Fig. 4B, the users whose posts are distributed in two adjacent ranks have the largest distribution in the 3rd and 4th categories, but the percentage difference between the three cases is not large. According to Fig. 4C, the users whose posts are distributed in three adjacent ranks are concentrated in type 234, which further indicates that the topic richness is positively correlated with the level of rank.

## CONCLUSION

Aiming at the importance and necessity of social media topic richness prediction in the field of information processing, this article proposes an English topic richness prediction algorithm based on the transformer. Firstly, the data that need to be considered are sorted out and extracted after analyzing the characteristics of Twitter. Secondly, feature fusion is adopted to mine, extract and construct the features of Twitter blogs and users, including blog features, topic features and user features, which are fused into multimodal features. Finally, the fused features are combined with the Transformer model for training and learning. The user features are constructed and utilized twice to construct a taxonomized topic richness prediction architecture, which can realize the prediction of Twitter topic richness. The proposed algorithm confirms the high influence of user features on richness prediction and achieves superior results on the dataset of Twitter topic richness. In practical applications, the proposed algorithm can be used to reveal personal preferences and public concerns in social media and help to predict the trend of social public opinion. In upcoming research endeavours, we aim to delve even deeper into the exploration of user characteristics. This involves considering additional factors like social network relationships and user behaviour patterns. The objective is to enhance the algorithm's precision in capturing users' personalized interests and preferences.

### Funding

This work was supported by Princess Nourah bint Abdulrahman University Researchers Supporting Project number (PNURSP2024R54), Princess Nourah bint Abdulrahman University, Riyadh, Saudi Arabia. The funders had no role in study design, data collection and analysis, decision to publish, or preparation of the manuscript.

## Grant Disclosures

The following grant information was disclosed by the authors:
Princess Nourah bint Abdulrahman University Researchers Supporting Project:
PNURSP2024R54.
Princess Nourah bint Abdulrahman University, Riyadh, Saudi Arabia.

## Competing Interests

The authors declare that they have no competing interests.

## Author Contributions

- Jie Jiao conceived and designed the experiments, performed the experiments, performed the computation work, prepared figures and/or tables, and approved the final draft.
- Hanan Aljuaid conceived and designed the experiments, analyzed the data, authored or reviewed drafts of the article, and approved the final draft.

## Data Availability

The data is available in at Zenodo: Twitter Dataset. (2022). Twitter Dataset [Data set]. Zenodo. https://doi.org/10.5281/zenodo.7139621.

## Supplemental Information

Supplemental information for this article can be found online at http://dx.doi.org/10.7717/peerj-cs.1967#supplemental-information.

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
