# Peer review of "Research on the prediction of English topic richness in the context of multimedia data"

_PeerJ Computer Science, doi:10.7717/peerj-cs.1967_

## Round 0.1 · original submission · Major Revisions

Based on the reviewers’ comments, you may resubmit the revised manuscript for further consideration. Please consider the reviewers’ comments carefully and submit a list of responses to the comments along with the revised manuscript.

**Language Note:** The review process has identified that the English language must be improved. PeerJ can provide language editing services - please contact us at copyediting@peerj.com for pricing (be sure to provide your manuscript number and title). Alternatively, you should make your own arrangements to improve the language quality and provide details in your response letter. – PeerJ Staff

Reviewer 1 ·

Basic reporting

No novelty in the paper. It looks like an implementation of a transformer using Twitter data. Therefore, the degree of advancement in Multimedia data analysis is very low. Additionally,

• Highlight tangible results in the abstract
• inadequate literature review
• The problem statement is unclear
• results are not compared with other state-of-the-art techniques
• inconsistent reference style

Experimental design

The rigorous evaluation of the proposed method is missing

Validity of the findings

The paper clearly fails to compare the results with the existing state-of-the-art techniques in multimedia data analysis

Additional comments

Authors carefully draft the article to attract the readers

Reviewer 2 ·

Basic reporting

This study has achieved significant progress in innovation, experimental design, result analysis, and other areas. However, there are some things that the author should pay attention to to improve the paper's overall quality.

You should improve the abstract by mentioning the main consequence of the proposed algorithm for predicting English subject richness.

Emphasize the innovation and uniqueness of your proposed Transformer-based algorithm. Clearly describe how this approach contributes to the discipline, particularly in terms of deep mining multimedia data.

Explain the process of feature fusion, including how blog features, topic features, and user features are mined, extracted, and built. This will assist readers grasp the many stages needed in preparing data for the algorithm.

Please ensure that all references are properly cited and formatted, following the designated academic style standard.

Experimental design

More information on how the fused characteristics are integrated into the model should be included in the section discussing the training and learning process with the Transformer model. A step-by-step overview of the process will help readers understand.

Give a full description of the taxonomized topic richness prediction architecture. Outline how user features are built and used twice, emphasising the architectural complexities that contribute to the algorithm's predictive ability.

Validity of the findings

Describe how your proposed algorithm can be used in the real world. Explain how the algorithm can show personal preferences and public concerns on social media and help predict social public opinion trends.

Briefly identify potential future work or study extensions, urging readers to think about the broader ramifications and potential developments in the topic.

There are a few errors in the technical terminology of the document, so it is strongly advised that a thorough proofread be performed.

Reviewer 3 ·

Basic reporting

I have noted some concerns that need to be incorporated in the revision. Addressing these issues mentioned below would help improve the quality of the paper. By incorporating these suggestions, your paper will not only be technically robust but also more accessible and engaging for a diverse audience interested in multimedia data mining and topic prediction on social media platforms.
The definition of this gap, which the author intends to fill in a better way than the solutions proposed by other researchers do not seem clear to the reader.
In the introduction, explicitly outline the practical significance of predicting topic richness in the context of public opinion monitoring and data discourse power competition. This will immediately engage readers and underscore the relevance of your work.

Experimental design

When discussing the analysis of Twitter characteristics, provide more specific details on the unique aspects of Twitter data that are considered. This could include the volume of tweets, user engagement patterns, or any other distinctive features.

Clarify the concept of multi-modal features, explaining how the fusion of blog, topic, and user features enhances the algorithm's ability to predict topic richness. Provide insights into the rationale behind combining these diverse features.

Emphasize the importance of user features in the proposed algorithm and elucidate why they play a pivotal role in predicting topic richness. This could be reinforced by citing examples or showcasing specific instances where user features significantly impact predictions.

Validity of the findings

In the results section, provide a nuanced interpretation of the findings, explaining not only the effectiveness but also the implications of the algorithm's performance. Discuss any patterns or trends observed in the dataset.

Consider integrating visual aids, such as flowcharts or diagrams, to illustrate the key steps and components of your proposed algorithm. Visual representations can significantly enhance the clarity of complex processes.

---

## Round 0.2 · accepted · Accept

Congratulations, the paper revisions are satisfactory and the reviewers have recommended the accept decision.

Reviewer 2 ·

Basic reporting

I recently had the opportunity to review the article, and I am pleased to report that the authors have successfully addressed all the concerns raised in the previous review. The diligence and commitment exhibited by the authors in resolving these observations have significantly enhanced the overall quality and credibility of the article.

The authors have provided a more detailed and transparent explanation of their research design, addressing the previous concerns. This enhancement strengthens the validity of their findings and adds to the article's accessibility to a broader audience.

The paper now stands as a commendable contribution to the field, with the authors successfully resolving all concerns raised in the previous review. The enhanced methodology, improved presentation, and responsiveness to feedback underscore the authors' commitment to scholarly excellence.

Experimental design

.

Validity of the findings

.

Reviewer 3 ·

Basic reporting

no comment

Experimental design

no comment

Validity of the findings

no comment

Additional comments

no comment